# Effect of Cytomegalovirus Reactivation on Inflammatory Status and Mortality of Older COVID-19 Patients

**DOI:** 10.3390/ijms24076832

**Published:** 2023-04-06

**Authors:** Robertina Giacconi, Maurizio Cardelli, Francesco Piacenza, Elisa Pierpaoli, Elisabetta Farnocchia, MirKo Di Rosa, Anna Rita Bonfigli, Tiziana Casoli, Francesca Marchegiani, Fiorella Marcheselli, Rina Recchioni, Pierpaolo Stripoli, Roberta Galeazzi, Antonio Cherubini, Massimiliano Fedecostante, Riccardo Sarzani, Chiara Di Pentima, Piero Giordano, Roberto Antonicelli, Mauro Provinciali, Fabrizia Lattanzio

**Affiliations:** 1Advanced Technology Center for Aging Research, IRCCS INRCA, Via Birarelli 8, 60121 Ancona, Italy; m.cardelli@inrca.it (M.C.); f.piacenza@inrca.it (F.P.); e.pierpaoli@inrca.it (E.P.); e.farnocchia@inrca.it (E.F.); m.provinciali@inrca.it (M.P.); 2Unit of Geriatric Pharmacoepidemiology and Biostatistics, IRCCS INRCA, 60124 Ancona, Italy; 3Scientific Direction, IRCCS INRCA, 60124 Ancona, Italyf.lattanzio@inrca.it (F.L.); 4Center for Neurobiology of Aging, IRCCS INRCA, 60121 Ancona, Italy; t.casoli@inrca.it; 5Center of Clinical Pathology and Innovative Therapy, IRCCS INRCA, 60121 Ancona, Italy; fr.marchegiani@inrca.it (F.M.); f.marcheselli@inrca.it (F.M.); r.recchioni@inrca.it (R.R.); p.stripoli@inrca.it (P.S.); 6Clinical Laboratory and Molecular Diagnostic, Italian National Research Center on Aging, IRCCS INRCA, 60127 Ancona, Italy; r.galeazzi@inrca.it; 7Geriatria, Accettazione Geriatrica e Centro di Ricerca per L’invecchiamento, IRCCS INRCA, 60127 Ancona, Italy; a.cherubini@inrca.it (A.C.); m.fedecostante@inrca.it (M.F.); 8Department of Clinical and Molecular Sciences, Università Politecnica delle Marche, 60126 Ancona, Italy; r.sarzani@inrca.it (R.S.); c.dipentima@inrca.it (C.D.P.); p.giordano@inrca.it (P.G.); 9Internal Medicine and Geriatrics, Italian National Research Centre on Aging, Hospital “U. Sestilli”, IRCCS INRCA, 60127 Ancona, Italy; 10Cardiology Unit, IRCCS INRCA, 60129 Ancona, Italy; r.antonicelli@inrca.it

**Keywords:** COVID-19 patients, cytomegalovirus, aging, inflammation, mortality

## Abstract

Herpesviridae reactivation such as cytomegalovirus (CMV) has been described in severe COVID-19 (COronaVIrusDisease-2019). This study aimed to understand if CMV reactivation in older COVID-19 patients is associated with increased inflammation and in-hospital mortality. In an observational single-center cohort study, 156 geriatric COVID-19 patients were screened for CMV reactivation by RT-PCR. Participants underwent a comprehensive clinical investigation that included medical history, functional evaluation, laboratory tests and cytokine assays (TNF-α, IFN-α, IL-6, IL-10) at hospital admission. In 19 (12.2%) of 156 COVID-19 patients, CMV reactivation was detected. Multivariate Cox regression models showed that in-hospital mortality significantly increased among CMV positive patients younger than 87 years (HR: 9.94, 95% CI: 1.66–59.50). Other factors associated with in-hospital mortality were C-reactive protein (HR: 1.17, 95% CI: 1.05–1.30), neutrophil count (HR: 1.20, 95% CI: 1.01–1.42) and clinical frailty scale (HR:1.54, 95% CI: 1.04–2.28). In patients older than 87 years, neutrophil count (HR: 1.13, 95% CI: 1.05–1.21) and age (HR: 1.15, 95% CI: 1.01–1.31) were independently associated with in-hospital mortality. CMV reactivation was also correlated with increased IFN-α and TNF-α serum levels, but not with IL-6 and IL-10 serum changes. In conclusion, CMV reactivation was an independent risk factor for in-hospital mortality in COVID-19 patients younger than 87 years old, but not in nonagenarians.

## 1. Introduction

Herpesviridae reactivation is frequent in critically ill patients, including those affected by SARS-CoV-2 (severe acute respiratory syndrome coronavirus 2) [1,2,3,4]. After primary infection, herpesvirus can produce a latent infection, and once the balance between the host immunity and the latent virus is broken, especially in older immunosuppressed COVID-19 (coronavirus disease 2019 caused by SARS-CoV-2) patients, herpesvirus proliferation can cause serious illness, involving multiple organs (liver, lung and brain) [5,6,7].

Among herpesviruses, cytomegalovirus (CMV) has been associated with digestive symptoms, gastrointestinal complications and bleedings in COVID-19 patients [8,9]. Moreover, CMV reactivation has been related to various adverse clinical outcomes, such as prolonged mechanical ventilation (MV), extracorporeal membrane oxygenation (ECMO) duration, increased length of hospitalization, and mortality [10,11,12]. In older subjects, CMV infection influences the maturation and long-term composition of the immune repertoire [13]. Importantly, CMV causes clonal T cell proliferation with an expansion in the number of virus-specific effector and memory cells, and a substantial reduction in naïve T cell diversity [14,15]. This phenomenon may lead to a reduced immune response to novel viral infections such as SARS-CoV-2. Severe COVID-19 is correlated with a dysregulated innate immune response, including a limited and delayed type I interferon (IFN) response and hyperinflammation [16]. Evidence shows that higher CMV viremia is associated with increased IL-6 production [17] and could fuel the cytokine storm in COVID-19 patients [18].

Although many factors have been associated with a poor prognosis in patients with COVID-19 infection, including genetic and laboratory parameters [19,20,21], investigating the impact of CMV reactivation, especially in the oldest old patients could be extremely useful for early identification of subjects at risk of developing severe complications and death.

## 2. Results

The clinical characteristics of COVID-19 patients included in this study are summarized in Table 1. Among the 156 patients, 19 were CMV DNAemia positive (12.2%), and the mean age and gender distribution were similar between CMV positive and CMV negative patients. The incidence of CMV reactivation did not change in relation to the main comorbidities. No difference in CRP, neutrophil and lymphocyte counts, neutrophil/lymphocyte ratio, fibrinogen, D-dimer concentrations, clinical frailty scale, presence at admission of respiratory insufficiency, fever and cough was found between groups. CMV IgM antibodies were also tested. Overall, 21% (4/19) of patients with CMV DNAemia were CMV IgM positive, while among the patients without CMV DNAemia, 4.4% (4/137) resulted as CMV IgM positive (Appendix A). Moreover, CMV IgM positive patients showed similar IFN-α and TNF-α levels to CMV IgM negative patients, but lower than patients with CMV DNAemia (Appendix A).

Since previous evidence has shown that CMV T-cell changes were associated with mortality in older subjects under 85 years of age, but not in older ones [22], to investigate the possible association of CMV reactivation with in-hospital death, we stratified the patient cohort according to the median age. Table 2 reports the frequency distribution of CMV positive patients in relation to in-hospital death after age-stratification. In the group with age < 87 years (mean age 79.9 ± 5.7), the in-hospital mortality was higher in patients with CMV reactivation than without reactivation (50% vs. 13.2%; *p* < 0.01).

Cox regression analysis indicated that the risk of a poor prognosis in patients younger than 87 years with CMV viremia was 9.9 times higher than that in patients without CMV reactivation (*p* < 0.05) (Table 3). Other predictors of in-hospital death were CRP, neutrophil count and frailty status (CSF). In patients older than 87 years (mean age 90.7 ± 3.2), age and neutrophil count were the only factors significantly associated with increased mortality. In-hospital death was not associated with the number of patients’ comorbidities in either age groups.

Accordingly, survival analysis indicated that patients with CMV reactivation and younger than 87 years had a lower survival than CMV negative patients, while no difference was observed in nonagenarian patients (over 87 years of age) (Figure 1).

Cytomegalovirus (CMV); chronic obstructive pulmonary disease (COPD); ischemic heart disease (IHD); atrial fibrillation (AF); chronic kidney disease (CKD), C-reactive protein (CRP), neutrophil/lymphocyte ratio (NLR); clinical frailty scale (CSF); and interquartile range (IQR).

Kaplan–Meier plots compared in-hospital mortality between CMV positive (green line) and CMV negative patients (blue line). Panel A: Patients younger than 87 years with CMV reactivation had a lower survival than CMV negative ones. Panel B: CMV reactivation was not a predictor of in-hospital death in nonagenarian patients.

IFN-α and TNF-α serum levels were higher in CMV positive patients than CMV negative ones. No difference was observed for IL-6 and IL-10 serum concentrations. Data were analyzed in the whole patient cohort.

ANCOVA analysis correcting for age and sex. * *p* < 0.05; ** *p* < 0.01.

## 3. Discussion

COVID-19 is characterized by a mild to severe respiratory illness, which is influenced by age and comorbidities [23,24,25]. Older patients, presenting a chronic pro-inflammatory status and immunosenescence, have an increased risk to develop COVID-19 severe outcomes including mortality [26]. T cells have a crucial role in controlling viral infections. CMV seroprevalence increases with age, reaching 85–90% by 75–80 years [27,28]. In the elderly, CMV causes clonal T cell proliferation, and reduction in naïve T cell diversity, which in turn may lead to reduced immune responsiveness to novel viral infections such as SARS-CoV-2, making older people particularly susceptible to severe COVID-19 [29].

On the other hand, it has also been observed that the reactivation of endogenous latent viruses such as HSV and CMV is quite common in critically ill patients and immunosuppressed patients [30]. CMV reactivation has been associated with an increased risk of hospitalization due to SARS-CoV-2 infection [31]. In this study, we found that 12% of elderly COVID-19 patients presented CMV reactivation, in agreement with previous studies [4,32]. Moreover, CMV positive patients younger than 87 years had a higher risk of in-hospital mortality than those without CMV reactivation. Consistent with other investigations, we also found that CRP, neutrophil count and frailty status were significantly associated with in-hospital death [33,34,35], whereas in patients older than 87 years, who were more susceptible to mortality from SARS-CoV-2 infection with respect to the younger group (45% vs. 17%), CMV DNAemia was not associated with death. In accordance with previous evidence, in the oldest group the main factors related to in-hospital mortality were age and an increased neutrophil count [34,35,36]. The lack of association between CMV reactivation and in-hospital mortality in the oldest patients could seem unexpected, because of evidence from longitudinal studies, where nonagenarians show CMV T-cell changes (oligoclonal, memory T cells, a reduction in the naïve T cells), in turn associated with frailty and increased mortality [14,37]. However, our results could be partly explained by the findings of Johnstone et al. [22], who demonstrated that CMV-reactive CD4+ T-cells were predictive of mortality within 1 year in subjects aged 65 to 84, but not in the oldest ones (aged 85–104 years). However, due to the limited number of CMV positive nonagenarians, our finding should be confirmed in a larger population.

It is still unclear whether CMV reactivation occurs in the context of severe SARS-CoV-2 infection or whether ongoing CMV infection influences the pathogenesis of COVID-19. CMV IgM detection revealed that 21% (4/19) of patients with CMV DNAemia were CMV IgM positive, while among patients without CMV DNAemia, 4.4% (4/137) resulted as CMV IgM positive. However, it is likely that in very old patients IgM positivity does not indicate primary CMV infection but a condition of immunosuppression likely due to high-dose corticosteroid treatment [38].

However, different conditions associated with SARS-CoV-2 infection such as lymphocyte depletion, glucocorticoid therapy and the advanced age are all factors favoring CMV reactivation. In our cohort, glucocorticoid treatment was not associated with the risk of CMV-infection (8.3% in CMV positive vs. 12.9% in CMV negative patients, *p* = 0.531 Pearson Chi-square test). Moreover, even if lymphocyte count was not significantly different between CMV negative and CMV positive patients, a marked reduction of the total number of NK and CD8+ T cells in patients with SARS-CoV-2 infection has been demonstrated, a condition that may lead to immunosuppression favoring CMV reactivation [39].

It has been found that CMV replication predominates in the lung, a major reservoir for CMV, and local reactivation may cause lung injury and hyperinflammation, resulting in complications in critically ill patients [38,40]. Similarly to SARS-CoV-2 virus, CMV induces the production of cytokines and chemokines that can exacerbate inflammation, contributing to a cytokine storm in COVID-19 patients [41,42,43]. In this regard, we found increased TNF-α serum levels in CMV positive patients, although no difference in IL-6 and IL-10 serum concentrations was present between CMV positive and CMV negative patients.

Interestingly, CMV positive patients also had increased IFN-α serum levels that might be a response to CMV reactivation to counteract virus replication [44].

Type I interferons (IFNs) have emerged as crucial contributors to the immune response against a SARS-CoV-2 infection [45,46]. Previous findings suggested that an impaired IFN-I response could be an important contributor to the disease severity in SARS-CoV-2 infection [45,46,47]. However, recent evidence shows that serum IFN-I levels in the early phase of SARS-CoV-2 infection are increased in patients who develop hypoxemic respiratory failure [48]. In our study, the levels of cytokines in serum were measured within one week after symptom onset, therefore the increased early IFN-α response in CMV positive patients could contribute to disease severity.

This study has some limitations, in particular the retrospective design, the small sample size of our cohort with a reduced number of CMV positive patients and the evaluation of a single outcome, i.e., in-hospital mortality. Furthermore, we did not evaluate the effect of treatments performed during admission, which might influence patients’ evolution.

Therefore, the study results need to be confirmed in a larger patient cohort.

In summary, older COVID-19 patients may present CMV reactivation, which could aggravate the disease course, influencing IFN-α and TNF-α production and increasing the risk of in-hospital death in elderly octogenarian patients. However, to better understand how CMV may influence the severity of illness and whether CMV specific antiviral treatment may improve the prognosis in older patients with CMV reactivation, further investigations are needed.

## 4. Materials and Methods

### 4.1. Patients Recruitment

This study includes 156 patients with COVID-19, who were recruited between 1 October 2020 and 24 June 2021 in the framework of the Report-Age COVID-19 project, an observational study conducted at the Italian National Center on Aging (IRCCS INRCA). The mean age of patients was 85.4 ± 7.1. Nasopharyngeal and throat swab samples were obtained at admission from all patients who were tested using the real-time polymerase chain reaction (RT-PCR) assay to identify SARS-CoV-2 infection. All patients were unvaccinated against SARS-CoV-2. No patients underwent invasive mechanical ventilation, but were supported with non-invasive oxygen therapy. The study was approved by the Ethics Committee of IRCCS INRCA (number CE-INRCA-20008) and registered under the ClinicalTrials.gov database (reference number NCT04348396). All patients signed informed consent. All study protocols were performed in alignment with local and international guidelines and regulations, and the research has been conducted in accordance with the Declaration of Helsinki.

### 4.2. Data Collection

The clinical records and laboratory data of each patient were obtained from the electronic medical system and were anonymized before release. Clinical data included symptoms and signs of infection such as fever, cough, dyspnea, diarrhea, nausea, and vomitting. Frailty was graded according to the Rockwood Clinical Frailty Scale (CFS) [49]. The CFS is an ordinal scale that ranks frailty from 1 to 9 (from being very fit to severely frail and terminally ill), with higher scores indicating progressively higher degrees of frailty.

Comorbidities were assessed including hypertension, diabetes mellitus, chronic obstructive pulmonary disease, ischemic heart disease, atrial fibrillation, dementia and chronic kidney disease. Laboratory parameters at admission, including those for complete blood count, serum concentrations of C-reactive protein (CRP), and D-dimer, were tested in the IRCCS INRCA laboratory using standardized and certified procedures.

### 4.3. CMV Detection

Total viral DNA was extracted from 140 µL of plasma using QIAamp Viral RNA Mini Kit (QIAGEN, Milan, Italy) and following the manufacturer’s protocol.

Real-time quantitative PCR was performed using the following primers for CMV detection:

Forward 5′-CAGTCCCGAGACMGTGAGAC-3′;

Reverse 5′-TGAACATCCCCAGCATCAACG-3′.

A total volume of 15 μL was used, with 0.3 µM of each primer, 7.5 μL of iTaq Universal SYBR Green Supermix (Biorad, Milan, Italy) and 3 μL of DNA for CMV assay. All reactions were performed on Rotor-Gene Q (QIAGEN, Milan, Italy), with the following conditions: 3 min at 95 °C, 45 cycles of 15 s at 95 °C, and 30 s at 60 °C. Positive control DNA (AcroMetrix CMV High Controls) and negative template controls were included on each assay.

All samples were run on duplicate arrays, and discordant results run a third time. The plasma sample was considered positive for the CMV DNAemia when detected with a cycle threshold (CT) < 38.

### 4.4. Detection of IgM Antibodies to CMV

The CMV IgM were measured in serum samples with LIAISON^®^ CMV IgM II chemiluminescent immunoassay (Diasorin, Vercelli, Italy) following the manufacturer’s protocol. IgM were equivocal for values ≥ 18 U/mL and <22 U/mL or positive for values ≥ 22 U/mL.

### 4.5. Cytokine Determination

Serum IL-6, IL-10, TNF-α, and IFN-α were measured by using the high-sensitivity ProQuantum qPCR immunoassays (Thermo Fisher, Waltham, MA, USA) measured on the Aria Mix real-time PCR system (Agilent, Santa Clara, CA, USA) following the manufacturer protocol. Each sample was assayed in duplicate.

### 4.6. Statistical Analyses

Log transformation of the variables was carried out if they were not normally distributed as assessed by the Kolmogorov–Smirnov test. Patient characteristics were reported as mean ± standard error of the mean (SEM) or median values and interquartile range (IQR) for continuous variables, or absolute frequency and percentages for categorical variables. Differences between groups were analyzed by t test or Mann–Whitney test for continuous variables and Pearson’s χ^2^ test for categorical variables. ANOVA (after correction for age and sex) was used to evaluate differences in inflammatory and coagulation markers, between patients with or without CMV reactivation. The relationship between CMV reactivation and mortality was investigated by Kaplan–Meier and Cox regression analyses.

The main factors included in the Cox regression model were age, sex, disease count, CRP, CMV, neutrophil count and clinical frailty scale.

Disease count represented all diseases included in our dataset, in particular: hypertension, diabetes, chronic obstructive pulmonary disease, asthma, ischemic heart disease, dementia, atrial fibrillation and chronic kidney disease.

Statistical significance was set to 2-sided *p* < 0.05.

Statistical analyses were performed using SPSS (Version 27.0.1.0, IBM, Amenk, NY, USA).

## Figures and Tables

**Figure 1 ijms-24-06832-f001:**
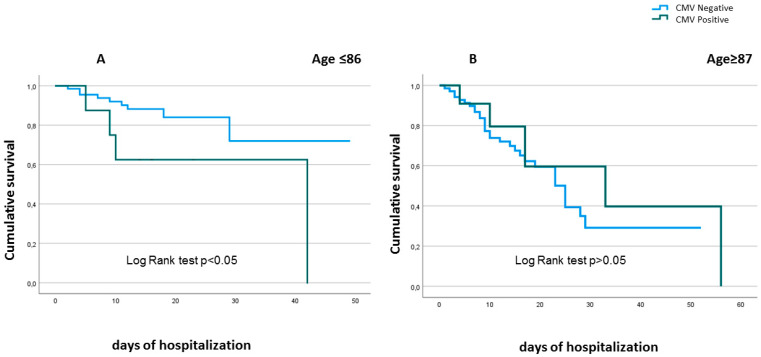
Kaplan–Meier survival estimates according to CMV reactivation in old COVID-19 patients stratified by age. Kaplan-Meier plots comparing in-hospital mortality between CMV positive (green line) and CMV negative patients (blue line). Panel A: Patients younger than 87 years with CMV reactivation had a lower survival than CMV negative ones. Panel B: CMV reactivation was not a predictor of in-hospital death in nonagenarian pa-tients.CMV reactivation was also associated with increased IFN-α and TNF-α serum levels, while no difference was observed for IL-6 and IL-10 serum concentrations in the whole patient cohort (Figure 2).

**Figure 2 ijms-24-06832-f002:**
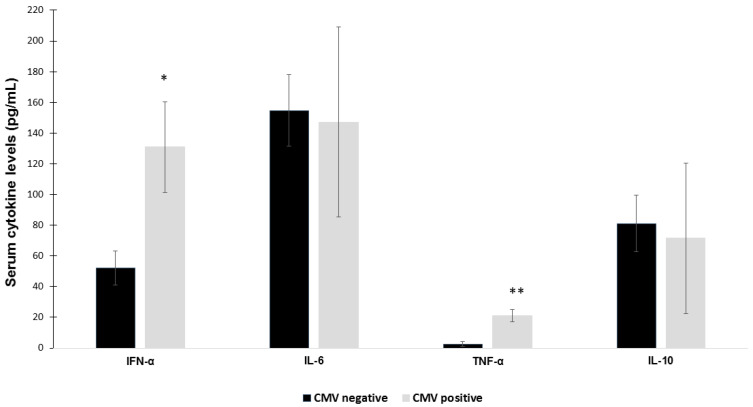
Serum cytokine concentrations in COVID-19 patients according to CMV reactivation. IFN-α and TNF-α serum levels were higher in CMV positive patients than CMV negative ones. No difference was observed for IL-6 and IL-10 serum concentrations. Data were analyzed in the in the whole patient cohort. ANCOVA analysis correcting for age and sex * *p* < 0.05; ** *p* < 0.01.

**Table 1 ijms-24-06832-t001:** Baseline clinical patient characteristics.

	Non-CMV Reactivation *n* = 137	CMV Reactivation *n* = 19	*p* Value
Age, mean ± SEM	85.22 ± 0.61	87.26 ± 1.64	0.245
Male sex, *n*(%)	54 (39.7%)	5 (25.0%)	0.199
Hypertension, *n* (%)	97 (71.0%)	12 (63.1%)	0.269
Diabetes, *n* (%)	33 (24.1%)	4 (21.0%)	0.496
COPD, *n* (%)	20 (14.6%)	2 (10.5%)	0.770
IHD, *n* (%)	17 (12.4%)	4 (21.0%)	0.300
AF, *n* (%)	36 (26.2%)	9 (47.0%)	0.057
Alzheimer, *n* (%)	14 (10.2%)	2 (10.5%)	0.967
CKD, *n* (%)	35 (25.5%)	4 (21.0%)	0.671
Respiratory insufficiency, *n* (%)	100 (73.0%)	14 (73.7%)	0.949
Fever, *n* (%)	66 (48.2%)	11 (57.8%)	0.427
Cough, *n* (%)	33 (24.0%)	5 (25.0%)	0.832
CRP (mg/L), mean ± SEM	5.41 ± 0.51	5.24 ± 1.33	0.821
Lymphocytes (103/µL), mean ± SEM	1.28 ± 0.07	1.07 ± 0.18	0.248
Neutrophils (103/µL), mean ± SEM	7.16 ± 0.38	8.61 ± 0.99	0.235
NLR, mean ± SEM	8.2 ± 0.2	10.9 ± 2.1	0.099
Fibrinogen (mg/dL), mean ± SEM	449.3 ± 12.1	456.8 ± 30.3	0.986
D-dimer (ng/mL), mean ± SEM	2005.6 ± 255.6	1903.3 ± 686.9	0.572
CFS Score, median (IQR)	7 (4–8)	7 (5–8)	0.916

**Table 2 ijms-24-06832-t002:** Frequency distribution of CMV positive patients in relation to in-hospital death after age-stratification.

		CMV Negative	CMV Positive
**Age ≤ 86**	Survivors, *n* (%)	59 (86.8%)	4 (50.0%)
	Deceased, *n* (%)	9 (13.2%)	4 (50.0%) *
**Age ≥ 87**	Survivors, *n* (%)	38 (55.1%)	6 (54.5%)
	Deceased, *n* (%)	31 (44.9%)	5 (45.5%)

* Pearson’s χ^2^ = 6823, *p* = 0.009.

**Table 3 ijms-24-06832-t003:** Multivariable cox regression analysis of associated factors with in-hospital mortality in COVID-19 patients stratified by age.

		HR	95% CI	*p* Value
**≤86 age yrs**	**age**	1.16	0.96–1.40	0.125
	**sex**	0.626	0.14–2.78	0.538
	**disease count**	0.88	0.55–1.40	0.598
	**CRP**	1.17	1.05–1.30	0.005
	**CMV**	9.94	1.66–59.50	0.012
	**Neutrophil count**	1.20	1.01–1.42	0.038
	**CFS**	1.54	1.04–2.28	0.032
**≥87 age yrs**	**age**	1.15	1.01–1.31	0.031
	**sex**	1.67	0.69–4.02	0.254
	**disease count**	1.32	0.99–1.77	0.056
	**CRP**	1.068	0.99–1.15	0.073
	**CMV**	0.394	0.11–1.38	0.146
	**Neutrophil count**	1.13	1.05–1.21	0.001
	**CFS**	1.01	0.80–1.28	0.916

C-reactive protein (CRP); cytomegalovirus (CMV); and clinical frailty scale (CSF).

## Data Availability

The datasets analyzed in this study are available from the corresponding author upon reasonable request.

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
