# Peer review of "Effect of Cytomegalovirus Reactivation on Inflammatory Status and Mortality of Older COVID-19 Patients"

_ijms, 2023, doi:10.3390/ijms24076832_

Round 1

Reviewer 1 Report

Since the CMV has been previously described in severe COVID-19. The present study revealed that the older COVID-19 patients may present CMV reactivation which could aggravate the disease course, influencing IFN-α and TNF-α production and increasing the risk of in-hospital death in elderly octogenarian patients. Despite it is an interesting manuscript. There are some concerns that need to be further investigated. 

1. Previously, one third of patients (about 70%) in intensive care reactivate CMV, which doubles their mortality rate; in this study, how many COVID-19 patients reactivate latent CMV to complicate their diseases and enhance their mortality rate? not well described

2. A  larger patient cohort needed to perform to verify their data

3. Methods for CMV testing should include testing for CMV IgM, and preferably examination of a tissue sample by immunostaining for CMV, in comparion with and without COVID-19

Reviewer 2 Report

This manuscript exploited the relationship of cytomegalovirus (CMV) reactivation with the inflammatory status and the result of older patients infected with COVID-19, and found CMV reactivation is an independent risk factor. These results would be value of guide for the therapy of COVID-19 older patients.   

The manuscript is interesting. However, a few comments should be addressed to improve the manuscript.

Comments:

1. Line 45,46: The format of comma seems not to be correct.

2.Line 81, 109…: the “P” should be italicized.

3. Line 107, 110: Table 2 and Table 3 should be three-line grid .

4. Line 113: This figure is loss of Y axis.

Round 2

Reviewer 1 Report

Thanks for the revised manuscript. There is no further comment. It can be now acceptable!